# “Cold” Somatostatin Analogs in Neuroendocrine Neoplasms: Decoding Mechanisms, Overcoming Resistance, and Shaping the Future of Therapy

**DOI:** 10.3390/cells14040245

**Published:** 2025-02-09

**Authors:** Sara Massironi, Manuela Albertelli, Iderina Hasballa, Piero Paravani, Diego Ferone, Antongiulio Faggiano, Silvio Danese

**Affiliations:** 1Faculty of Medicine and Surgery, Vita e Salute San Raffaele University, Via Olgettina, 20132 Milan, Italy; sdanese@hotmail.it; 2Gastroenterology Unit, Istituti Ospedalieri Bergamaschi, 24046 Bergamo, Italy; 3Endocrinology, Department of Internal Medicine and Medical Specialties (DiMI), University of Genova, 16132 Genova, Italyiderina.hasballa@unige.it (I.H.); diego.ferone@unige.it (D.F.); 4Endocrinology Unit, IRCCS Ospedale Policlinico San Martino, 16132 Genova, Italy; 5Unit of Endocrinology, Department of Clinical and Molecular Medicine, ENETS Center of Excellence, Sant’Andrea Hospital, Sapienza University, 00189 Rome, Italy; piero.paravani@uniroma1.it (P.P.); antongiulio.faggiano@uniroma1.it (A.F.); 6Gastroenterology and Endoscopy, IRCCS San Raffaele Hospital, 20132 Milan, Italy

**Keywords:** somatostatin analogs (SSAs), somatostatin receptors (SSTRs), neuroendocrine neoplasms (NENs), peptide receptor radionuclide therapy (PRRT), high-dose SSA therapy

## Abstract

Background. Neuroendocrine neoplasms (NENs) represent a heterogeneous group of tumors that pose significant therapeutic challenges due to their potential for progression, metastasis, and hormonal syndromes. Somatostatin analogs (SSAs) have emerged as a cornerstone in NEN treatment, offering both antisecretory and antiproliferative effects by targeting somatostatin receptors (SSTRs). Despite their proven efficacy, intrinsic and acquired resistance mechanisms, including receptor downregulation, tumor heterogeneity, and microenvironmental influences, limit their long-term effectiveness. Recent advances, including high-dose SSA regimens and novel formulations, have aimed to optimize their therapeutic utility and address these limitations. Body of the review. This review explores the cellular and molecular mechanisms underlying the antitumor effects of SSAs, including receptor-mediated signaling pathways, cell cycle arrest, apoptosis induction, and antiangiogenesis. The role of SSAs in combination therapies with mTOR inhibitors and peptide receptor radionuclide therapy (PRRT) is analyzed, emphasizing their synergistic potential. Key clinical trials, such as RADIANT-2, EVERLAR, and NETTER-1, support the efficacy of these approaches, demonstrating improved outcomes when SSAs are combined with targeted agents or radiolabeled therapies. Emerging strategies include high-dose SSA regimens, particularly in progressive cases with low Ki67 indices. Finally, novel formulations, including oral octreotide, paltusotine, and subcutaneous depot formulations like CAM2029, offer improved pharmacokinetics, bioavailability, and patient adherence. Ongoing clinical trials, including SORENTO, further evaluate their efficacy and safety profiles. Conclusions. This paper provides a comprehensive analysis of the cellular and molecular mechanisms of SSAs. SSAs remain integral to the management of NENs, providing effective tumor stabilization and symptom control. However, resistance mechanisms and tumor heterogeneity necessitate innovative approaches, including high-dose regimens, combination strategies, and next-generation formulations. Future research should focus on refining these strategies to optimize patient outcomes, enhance long-term efficacy, and expand the therapeutic landscape for NENs.

## 1. Introduction

Neuroendocrine neoplasms (NENs) are a heterogeneous group of tumors derived from neuroendocrine cells distributed throughout the body, most commonly in the gastrointestinal tract, pancreas, and lungs [1,2,3]. The incidence of neuroendocrine tumors (NETs) has been steadily increasing over the past few decades [4], with a noticeable shift in demographics [5,6,7]. Historically, NETs were more commonly observed in individuals in their sixth or seventh decade of life. However, recent data indicate that younger populations are now being diagnosed with these tumors, reflecting a concerning trend. This shift underscores the need for ongoing research and adaptation of diagnostic and therapeutic strategies.

According to the latest WHO classification, NENs are broadly categorized into well-differentiated neuroendocrine tumors (NETs) and poorly differentiated neuroendocrine carcinomas (NECs) [8,9]. NETs are graded into Grade 1 (G1), Grade 2 (G2), and Grade 3 (G3) based on mitotic count and the Ki-67 proliferation index, reflecting their varying degrees of aggressiveness [10]. NECs, on the other hand, are universally classified as G3 and exhibit high-grade, poorly differentiated morphology. Although many NENs exhibit slow progression, their potential for metastasis and hormone-related syndromes poses significant clinical challenges [11,12,13]. Effective management requires a multidisciplinary approach that integrates symptom control and tumor stabilization.

Somatostatin and its analogs represent an ideal therapeutic agent for NENs as they combine both antisecretory and antiproliferative properties, achieving their effects through binding to somatostatin receptors (SSTRs) expressed on neoplastic neuroendocrine cells [14,15]. Indeed, somatostatin analogs (SSAs), such as octreotide and lanreotide, beyond their regulatory secretion role, have also demonstrated antiproliferative effects [16,17,18,19] in randomized clinical trials and have indeed revolutionized NEN management [16,17,18,19]. These agents, designed to mimic endogenous somatostatin, exhibit enhanced receptor affinity and longer half-life [19]. Despite their well-established clinical utility, the precise cellular and molecular mechanisms underpinning the antiproliferative effects of SSAs are still being elucidated, with receptor downregulation, tumor heterogeneity, and microenvironmental influences posing therapeutic challenges [20,21,22].

This review examines the mechanistic insights into SSAs and discusses emerging therapeutic strategies—such as combination treatments, high-dose regimens, and novel drug delivery systems—that aim to address current limitations and enhance the personalized management of NENs [23].

## 2. Normal Distribution and Function of Somatostatin Receptors and Their Role in Neuroendocrine Neoplasms

Somatostatin receptors (SSTRs) are widely expressed in normal tissues and organs, where they play crucial roles in regulating endocrine and exocrine functions, cell proliferation, and neurotransmission. The five SSTR subtypes (SSTR1-SSTR5) have different but sometimes overlapping distributions. SSTR1 is primarily found in the central nervous system (CNS), gastrointestinal (GI) tract, and kidneys, where it regulates neuronal signaling and GI motility. SSTR2 is ubiquitously expressed in the CNS, endocrine pancreas, and GI tract and is responsible for the inhibition of hormone secretion and proliferation. SSTR3 is found in the pancreas, lungs, and brain, where it is involved in cell cycle arrest and apoptosis. SSTR4 is mainly found in the brain and lungs, where it modulates neurotransmission. SSTR5 is localized in the pituitary gland, pancreas, and GI tract and plays a role in regulating hormone secretion (Table 1). These receptors mediate their physiological effects through inhibitory G-protein-coupled mechanisms, resulting in reduced cyclic AMP levels, ion channel modulation, and inhibition of protein synthesis.

SSTRs are pivotal in the biology and treatment of NENs. These receptors belong to the G-protein-coupled receptor (GPCR) family and are divided into five subtypes, SSTR1 to SSTR5 [24]. Among these, SSTR2 is the most prevalent and therapeutically relevant in NENs, particularly in well-differentiated tumors. SSTR5 is the next most commonly expressed receptor, while SSTR1, SSTR3, and SSTR4 are less frequently present. The density and distribution of SSTRs on the surface of tumor cells vary depending on tumor grade, origin, and differentiation status. Well-differentiated NENs, which are often associated with a better prognosis, generally exhibit higher receptor expression compared to poorly differentiated and aggressive forms of the disease [25,26].

SSTR expression is critical not only for tumor biology but also for clinical management. These receptors modulate endocrine and paracrine signaling and serve as molecular targets for SSAs, thereby making them the cornerstone in NEN therapy [19,27,28,29]. The overexpression of SSTR2 supports SSA efficacy in controlling hormonal syndromes and stabilizing tumor growth [26,29,30]. Importantly, the expression of this receptor is also the basis for diagnostic imaging using radiolabeled SSAs, enabling the functional visualization of NENs [31].

The advent of advanced imaging modalities such as Gallium-68 (Ga-68) DOTATATE positron emission tomography (PET)/computed tomography (CT) has further underscored the importance of SSTRs in clinical practice. This technique leverages the high affinity of radiolabeled SSAs for SSTR2 to provide highly sensitive and specific visualization of NENs [32], aiding in diagnosis, staging, treatment planning, and monitoring therapeutic response. Furthermore, imaging can identify patients most likely to benefit from peptide receptor radionuclide therapy (PRRT), offering a more tailored therapeutic approach. Recent advancements in imaging modalities have significantly improved the diagnostic accuracy for NENs [33]. While 68Ga-DOTATATE remains the most widely used imaging agent, offering excellent specificity and sensitivity for SSTR-positive tumors, emerging alternatives such as 18F-NOTA-Octreotide are gaining traction. This novel modality provides enhanced image resolution due to the shorter positron range of 18F and offers logistical advantages owing to the longer half-life of 18F compared to 68Ga. This allows centralized cyclotron production and wider distribution, overcoming some limitations associated with 68Ga-DOTATATE. Clinical studies have also demonstrated comparable, if not superior, diagnostic performance with 18F-NOTA-Octreotide, particularly in detecting smaller lesions or those with lower receptor density. The integration of this imaging technique into routine practice has the potential to refine diagnosis, staging, and treatment planning for NEN patients [34].

Despite their utility, the functional status of SSTRs is influenced by the tumor microenvironment and its interactions with other cellular and molecular components, including receptor internalization, desensitization, and heterogeneity in receptor expression [21]. These variations impact therapeutic outcomes, with well-differentiated NENs showing robust responses to SSAs, while poorly differentiated tumors with low or absent receptor expression often require alternative strategies [35,36].

Emerging research is shedding light on the complex regulatory mechanisms governing SSTR expression and signaling. Factors such as hypoxia, inflammation, and epigenetic modifications are being investigated for their roles in modulating receptor density and activity [37,38]. Such insights may lead to strategies that enhance receptor expression or reactivate signaling pathways, improving the efficacy of SSTR-targeted therapies [39]. For instance, combination approaches involving epigenetic modulators or agents that modify receptor recycling may help overcome resistance to SSAs [40]. As research continues to unravel the complexities of SSTR biology, the findings promise to refine current therapeutic paradigms and inspire novel interventions, ultimately advancing the personalized management of patients with NENs.

## 3. Cellular Mechanisms of SSA Action and Antiproliferative Effects of SSAs

SSAs, as already stated, exert their therapeutic effects in NENs primarily through their interaction with SSTRs expressed on tumor cells. Among these, SSTR2 and, to a lesser extent, SSTR5 serve as molecular gateways, mediating both hormonal suppression and antiproliferative actions [41] (Table 2). Upon binding, SSAs induce conformational changes that couple with G-proteins, inhibiting adenylate cyclase and reducing cyclic adenosine monophosphate (cAMP) levels, thereby suppressing hormone secretion and growth [42,43,44].

However, beyond cAMP modulation, SSAs exert additional antiproliferative effects through multiple mechanisms [45,46,47] (Figure 1).

One major mechanism involves the modulation of key signaling pathways activating phosphotyrosine phosphatases (PTPs), which counteract oncogenic tyrosine kinase signaling, which are often upregulated in cancer and promote tumor proliferation [48]. By activating PTPs, SSAs suppress mitogenic pathways such as the mitogen-activated protein kinase (MAPK) and extracellular signal-regulated kinase (ERK) pathways [49].

Another primary antiproliferative mechanism of SSAs is their ability to induce cell cycle arrest, particularly at the G1 phase of the cell cycle [46]. The cell cycle is tightly regulated by cyclins and cyclin-dependent kinases (CDKs), which drive progression through its phases [50]. By downregulating cyclin D1 and inhibiting CDK4/CDK6 activity, SSAs prevent the phosphorylation of the retinoblastoma protein (Rb) [45]. In its hypophosphorylated state, Rb suppresses E2F transcription factors, blocking gene expression required for S-phase entry and DNA replication [46,51]. Moreover, the impact of SSAs on the cell cycle is closely linked to their modulation of signaling pathways downstream of SSTR activation. For instance, the inhibition of the MAPK/ERK and PI3K/AKT pathways by SSAs disrupts the transcriptional regulation of cyclin genes, further reinforcing their ability to arrest the cell cycle [52].

SSAs also promote apoptosis through a tightly regulated process. SSAs upregulate pro-apoptotic proteins such as Bax, a member of the Bcl-2 family, which promotes mitochondrial outer membrane permeabilization (MOMP) and the release of cytochrome c from the mitochondria. This release triggers the formation of the apoptosome, leading to the activation of caspase-9 and downstream effector caspases, such as caspase-3 [53,54]. On the other hand, SSAs downregulate anti-apoptotic proteins such as Bcl-2, which normally act to preserve mitochondrial integrity and prevent apoptosis. This shift in the Bax/Bcl-2 ratio disrupts mitochondrial stability, thereby favoring caspase activation and the execution of the apoptotic program [55]. These apoptotic effects are more pronounced in well-differentiated NENs with high SSTR density. The strong expression of SSTR2, in particular, enhances the sensitivity of tumor cells to SSA-induced apoptotic signaling. The dual actions of SSAs on cell cycle arrest and apoptosis contribute significantly to their antitumor efficacy. By halting tumor cell proliferation and promoting apoptosis, SSAs achieve a cytostatic effect that stabilizes disease progression, particularly in well-differentiated NENs [56]. Clinical studies have correlated high SSTR expression with improved tumor control and longer progression-free survival in patients treated with SSAs, highlighting the relevance of these mechanisms in therapeutic outcomes [57]. In preclinical models, SSA-induced apoptosis has been confirmed through markers such as increased caspase-3 activity and DNA fragmentation [58]. These findings align with observations in patient-derived tumor samples, where SSAs have been shown to reduce tumor viability and increase apoptotic cell death.

Moreover, SSAs influence critical signaling pathways such as the phosphoinositide 3-kinase (PI3K)/AKT/mammalian target of rapamycin (mTOR) pathway, a critical regulator of cell survival, metabolism, and growth [59,60,61]. This interaction is particularly relevant in the context of combination therapies, where SSAs are paired with mTOR inhibitors like everolimus to achieve synergistic therapeutic outcomes.

Finally, in addition to targeting proliferation and apoptosis, SSAs exert antiangiogenic effects by reducing the secretion of vascular endothelial growth factor (VEGF) and other proangiogenic mediators [62]. This limits the formation of new blood vessels, depriving tumors of nutrients and oxygen, thus impairing growth [63] and metastatic potential [64].

While these mechanisms collectively explain the clinical benefits of SSAs (Table 2), their efficacy is not uniform across all patients or tumor types. Variability in SSTR expression, receptor desensitization, and the interplay of other molecular pathways can influence the response to SSAs. Moreover, poorly differentiated NENs, which often lack significant SSTR expression, are less responsive to SSA therapy, highlighting the importance of patient selection based on receptor profiling.

## 4. Resistance Mechanisms and Limitations

Despite their established efficacy, SSAs are not universally effective in all NENs. A significant proportion of patients experience diminished responses over time or exhibit intrinsic resistance to these therapies [20]. The underlying mechanisms of resistance to SSAs are multifactorial, involving receptor dynamics, such as receptor internalization and degradation, the formation of homodimers and heterodimers, tumor heterogeneity, and the influence of the tumor microenvironment [39]. These limitations underscore the need to better understand resistance pathways and develop strategies to overcome them.

Moreover, while SSAs remain a cornerstone of therapy for NENs, their use is not without challenges. One notable complication associated with prolonged SSA therapy is the increased risk of gallstone formation [65,66], which results from the inhibitory effect of SSAs on gallbladder motility and bile secretion. This highlights the importance of monitoring patients on long-term SSA therapy for potential biliary complications and evaluating them for prophylactic or therapeutic cholecystectomy when necessary [67].

### 4.1. Downregulation of Receptors

One of the most well-documented mechanisms of resistance to SSAs is the downregulation or desensitization of SSTRs following prolonged exposure to the analogs [68]. When SSAs bind to SSTRs, the receptor–ligand complex undergoes internalization into the cell. While this process is initially reversible, chronic stimulation can result in reduced receptor recycling to the cell surface and eventual degradation. This leads to a decrease in receptor density, particularly of SSTR2, which is the primary target of most SSAs. The loss of functional receptors limits available binding sites, impairing the efficacy of SSAs. Studies have demonstrated that receptor downregulation is more pronounced in patients who have received long-term SSA therapy, which may explain the waning effectiveness observed in some cases. Treatment strategies such as dose escalation may temporarily restore responses during an initial phase of resistance. However, a secondary resistance phase often emerges, where increasing doses fail to elicit therapeutic effects [69,70]. This highlights the need for strategies to prevent receptor desensitization or promote receptor re-expression.

While most SSAs binding to SSTRs trigger receptor internalization, the degree of this process varies depending on the specific receptor subtype and ligand involved. However, not all ligands induce this effect. Notably, somatostatin receptor antagonists have been reported in the literature as binding to SSTRs without inducing internalization. These antagonists maintain the receptors on the cell surface, potentially enhancing their utility for therapeutic and diagnostic purposes. This distinct mechanism of action opens new avenues for improving SSTR-targeted approaches in NEN management [20].

### 4.2. Tumor Heterogeneity

Tumor heterogeneity poses a major challenge to the uniform efficacy of SSA therapy. SSTR expression varies widely between patients, among different tumor sites within the same patient, and even within individual tumors, or based on mitotic count and the Ki-67 index (Table 3).

Well-differentiated NENs (G1, G2, or even G3, according to the WHO classification) typically exhibit high SSTR density, particularly of SSTR2, making them more responsive to SSA therapy [71]. Conversely, poorly differentiated (NEC) and more aggressive NENs often exhibit reduced or absent SSTR expression, limiting SSA effectiveness [72,73,74]. In some cases, SSTR5 expression has been associated with the presence of metastases and a worse prognosis [75]. This heterogeneity is driven by the molecular and genetic diversity of NENs, including variations in gene expression, epigenetic modifications, and mutations affecting receptor signaling pathways. For example, SSTR expression loss has been linked to epigenetic silencing through promoter methylation in some NENs [76]. Additionally, the differential expression of SSTR subtypes can impact the efficacy of specific SSAs, as some analogs preferentially target certain receptor subtypes. SSA resistance mechanisms also differ significantly across NET site origin, influenced by variations in somatostatin receptor expression, receptor desensitization, and downstream signaling alterations. In GI and pancreatic NETs, lower SSTR expression or mutations in receptor pathways contribute to reduced SSA responsiveness. In contrast, pit-NETs often retain higher SSTR density, particularly SSTR2, leading to better treatment outcomes.

From a clinical perspective, this variability may necessitate careful patient selection and tailoring of SSA therapy based on receptor profiling [77]. Advanced imaging techniques, such as Ga-68 DOTATATE PET/CT, provide crucial insights into SSTR expression, enabling the identification of patients most likely to benefit from SSA treatment [74]. Recently, 18F-NOTA-Octreotide has emerged as a promising technique, offering enhanced resolution and logistical advantages over 68Ga-based tracers, further refining the diagnostic and therapeutic landscape for NENs [78].

These imaging modalities detect SSTRs subtypes and assess spatial and temporal heterogeneity, improving treatment monitoring and response evaluation. However, even in patients with high SSTR expression, resistant tumor subpopulations may persist, emphasizing the need for strategies to overcome this variability.

### 4.3. Role of the Tumor Microenvironment

The tumor microenvironment (TME) plays a crucial role in modulating SSA responsiveness. NENs exist within a complex microenvironment composed of stromal cells, immune cells, blood vessels, and extracellular matrix components, all of which interact with tumor cells to influence their behavior. Several factors within the TME can diminish the efficacy of SSAs, including hypoxia, immune evasion, and changes in the local cytokine milieu. Hypoxia, a common hallmark of solid tumors, has been shown to downregulate SSTR expression and impair SSA binding [79]. Under low oxygen conditions, tumor cells undergo metabolic and transcriptional reprogramming that can reduce receptor density and alter signaling pathways [80]. This hypoxic adaptation not only compromises the direct antiproliferative effects of SSAs but also promotes tumor aggressiveness and resistance to other therapies.

Immune evasion mechanisms within the TME further complicate SSA efficacy. NENs are often characterized by relatively low immunogenicity and an immunosuppressive microenvironment [81]. While SSAs have been reported to modulate immune responses in some settings, the immunosuppressive milieu of NENs may counteract these effects, limiting their ability to harness the immune system for therapeutic benefit [82].

Moreover, the dynamic interactions between tumor cells and stromal components can influence SSTR expression and signaling. For example, certain cytokines and growth factors secreted by stromal cells may downregulate SSTRs or activate alternative proliferative pathways that bypass SSA-mediated inhibition [83]. Understanding these interactions is critical for developing combination therapies that target both the tumor cells and their microenvironment.

## 5. Implications and Future Directions

The limitations of SSAs due to receptor downregulation, tumor heterogeneity, and microenvironmental influences underscore the need for innovative approaches to overcome resistance [77]. Potential strategies include the development of next-generation SSAs with enhanced receptor affinity and broader subtype specificity, combination therapies targeting complementary pathways, and interventions aimed at modifying the tumor microenvironment [23].

Mechanistic insights into SSAs also provide a framework for advancing their use in combination therapies and in the treatment of more challenging NENs. Ongoing research explores strategies to enhance the effectiveness of SSAs by targeting additional pathways or combining them with other therapies. For instance, combining cold SSAs with PRRT, angiogenesis inhibitors, chemotherapeutic agents, or immunotherapies could expand their utility in more aggressive NEN subtypes [23]. The potential of alpha emitters, such as 225Ac and 213Bi, to generate double-strand DNA breaks, leading to significantly enhanced cytotoxicity and tumor cell killing, represents an exciting area of research for enhancing SSA-based therapies [84,85]. Initial studies have shown promising results in combining alpha emitters with SSAs, paving the way for more potent and targeted therapeutic approaches, even in aggressive NENs [85]. However, further investigation is warranted.

Combining SSAs with epigenetic modulators such as DNA methyltransferase inhibitors could restore SSTR expression in tumors with epigenetically silenced receptors [86]. Similarly, targeting hypoxia or reprogramming the immune microenvironment may enhance the efficacy of SSAs in resistant tumors [79,80]. These approaches, coupled with advanced diagnostic techniques for patient stratification, hold the promise of expanding the therapeutic potential of SSAs and improving outcomes for patients with neuroendocrine neoplasms.

Additionally, novel formulations and delivery methods, such as high-dose SSAs, and subcutaneous or oral preparations, are being developed to optimize their pharmacological profiles and patient compliance.

### 5.1. Long-Term Use and Treatment Sequencing

One of the most pressing questions in SSA therapy is determining the optimal duration of use, particularly in patients who achieve partial or complete responses. While SSAs are often continued indefinitely in responders, the benefits of this approach versus stopping therapy after stabilization remain unclear.

A prospective, randomized study found no improvement in progression-free survival (PFS) or overall survival (OS) in patients with advanced, well-differentiated NENs who continued SSA therapy after achieving disease control with PRRT. These results were compared to a control group receiving only the best supportive care. The progressive disease (PD) rate was significantly higher in the SSA group (70%; 52/74) compared to the control group (46%; 19/41) (*p* = 0.01) [87].

In a comparative analysis of 15 prospective trials conducted by Mohamed et al., the addition of SSA as a maintenance strategy after targeted therapy (Everolimus, Sunitinib, Surufatinib, Lenvatinib, Nintedanib, Pazopanib, Axitinib) in refractory metastatic NETs was not associated with significant benefits in terms of survival or tumor control [88].

In contrast, promising results were documented by REMINET, a phase II/III, randomized, multicenter trial (NCT02288377), on continuing SSAs after stabilization or objective response induced by first-line treatment in aggressive and unresectable duodenal-pancreatic (DP) G1-G2 NETs. Fifty-three patients were enrolled and randomized 1:1 to lanreotide ATG 120 mg or placebo once monthly. At randomization, stable disease (SD) and partial response (PR) were observed in 81.1% and 18.9% of cases, respectively. Regarding first-line treatment, 96.8% received chemotherapy (CHT) (52.9% temozolomide-based, 18.7% dacarbazine-based, 13.3% streptozotocin-based, 11.3% oxaliplatin-based) and 3.8% received sunitinib. The 6-month PFS, the primary endpoint, was 73.1% (90% CI, 55.3–86.6) in the lanreotide arm, significantly higher than 54.2% (90% CI, 35.8–71.8) in the placebo arm. Median PFS was 19.4 months (95% CI, 7.6–32.6) in the lanreotide arm compared to 7.6 months (95% CI, 3.0–9.0) in the placebo arm. Median OS was not reached in patients receiving lanreotide, while in those on placebo, it was 41.9 months [89].

SSAs as maintenance or combination therapy with PRRT also demonstrated survival benefits in a retrospective study enrolling 168 patients with advanced G1-G2 GEP-NET. Eighty-one patients received PRRT only (group 1), whereas 87 received SSA combined with PRRT and/or continued after PRRT (group 2). Median PFS and OS rates were significantly higher in group 2 versus group 1, resulting in 48 months vs. 27 months (*p* = 0.012) and 91 months vs. 47 months (*p* < 0.001), respectively. Also, the imaging response rate was superior in group 2 (63.1%, clinical benefit rate—CBR 95.2%) compared to group 1 (40.0%, CBR 78.7%) among the 159 patients evaluated [90]. Additional evidence from the study of Sowa-Staszczak et al. reported prolonged survival with SSAs following PRRT (90Y/177Lu-DOTA-TATE) in 79 patients with advanced, well-differentiated NENs, with PFS, event-free survival (EFS), and OS of 39, 33, and 60 months, respectively. At 12 months of follow-up, 66% of patients presented SD, 21% PR, and 13% PD. The survival data of this study, specifically PFS and OS, were superior to prior findings of studies evaluating either SSA or PRRT in monotherapy [91].

For specific groups of NENs, retrospective studies have demonstrated prolonged PFS in patients with sporadic PanNENs smaller than 2 cm who were treated with SSAs compared to those under active surveillance [92]. This is further supported by the LARO-MEN1 study, a prospective and non-comparative trial, which assessed the efficacy of SSAs in MEN1 syndrome-associated PanNEN, highlighting their role in stabilizing disease even in hereditary conditions. Specifically, the preventive treatment with octreotide 10 mg/28 days i.m. in the eight enrolled patients with MEN1, panNET < 2 cm, and abnormal biomarkers at baseline (increased levels of glucagon in 7/8 and of somatostatin in 1/8) led to a reduction in tumor secretory activity, with normalization of PP, glucagon, and somatostatin values, as well as maintenance of disease stability [93]. Promising results were also observed by the retrospective study of Ramundo et al., which confirmed the efficacy of octreotide LAR i.m. in tumor control and hormonal response in early-stage MEN1 duodenal–pancreatic NETs < 2 cm. Besides the normalization of chromogranin A, insulin, and gastrin concentrations, an objective tumor response occurred in 10% of the cases, SD in 80% of the cases, and PD in 10% of the cases [94]. Also, subcutaneous octreotide was previously shown to be a safe and effective adjunct to surgery in patients with MEN1 GEP-NET and hypergastrinemia [95].

Future trials, such as the SAUNA trial, aim to clarify the role of SSA maintenance therapy in progressive, advanced GEP-NETs, stratified by Ki-67 and treatment type. The SAUNA trial (NCT05701241) is a multicenter, randomized phase IV trial, recruiting patients with non-functional, advanced GEP-NET in progression after first-line treatment with SSAs. Based on the second-line strategy (i.e., PRRT or targeted therapies), eligible subjects will be allocated to two substudies and subsequently randomized 1:1 to maintenance or withdrawal of SSAs (octreotide LAR 30 mg or lanreotide ATG 120 mg) according to the tumor site and Ki67 (<10%, G1 or low G2, or ≥10%, high G2). Primary endpoints include the difference in PFS, assessed on cross-sectional imaging according to RECIST 1.1 criteria, and the difference in time to deterioration (TTD) between patients maintaining or discontinuing SSAs within each substudy. This will be further explored by the ongoing interventional phase III SANO trial (NCT02705651), in which the benefits of SSAs compared to “no treatment” will be analyzed with regard to progression (tumor growth; development of new neuroendocrine tumors and regional/distant metastasis).

### 5.2. Combination Therapies and Synergistic Effects

Combining SSAs with other treatments offers the potential for enhanced efficacy, particularly in advanced or refractory cases.

#### 5.2.1. SSAs and mTOR Inhibitors

The combination of SSAs with mTOR inhibitors, such as everolimus, represents a promising strategy to enhance therapeutic outcomes in advanced NENs. This combination provides a dual mechanism of action that targets complementary pathways involved in tumor proliferation and survival, making it a viable strategy for refractory NETs [96]. Preclinical studies, specifically in pituitary and pancreatic adenocarcinoma cells, have demonstrated synergistic effects through dual inhibition of the PI3K/AKT/mTOR pathway. Specifically, activation of SSTR2 by SSAs suppresses PI3K activity, interfering with the p85 PI3K regulatory subunit, thereby amplifying the antiproliferative effects of mTOR inhibitors [97]. Moreover, the PI3K/AKT/mTOR pathway is often upregulated in aggressive NETs, contributing to resistance mechanisms. The combination therapy addresses this challenge by targeting distinct yet interconnected components of the pathway. In addition to direct cytostatic effects, this combination modulates the tumor microenvironment [98]. SSAs exhibit antiangiogenic properties by downregulating VEGF, while mTOR inhibitors disrupt tumor-associated stromal support and nutrient supply [99].

Clinical trials, such as the RADIANT series, have provided further evidence supporting this combination. The phase III RADIANT-2 trial, a double-blind, placebo-controlled randomized trial, investigated everolimus combined with octreotide LAR in patients with advanced, low- or intermediate-grade NETs and carcinoid syndrome, suggesting a synergistic effect, particularly in patients with advanced NENs associated with secretory symptoms. While the improvement did not always meet statistical significance in all the trials of the series (RADIANT-1-4), the RADIANT-2 trial demonstrated a significant improvement in median PFS (16.4 months vs. 11.3 months, hazard ratio 0.77, 95% CI 0.59–1.00; one-sided log-rank test *p* = 0.026) compared to octreotide LAR alone [100]. Similarly, the EVERLAR study, a phase II prospective single-arm trial, confirmed antitumor activity with everolimus (10 mg/day) and octreotide (30 mg/month) in patients with advanced, non-functioning GI-NETs. The 12-month and 24-month PFS rates were 62.3% (CI 95% 48–77%) and 43.6% (95% CI 29–58%), respectively, with a disease control rate of 58.1%. Notably, median overall survival (mOS) was not reached after 24 months of follow-up, highlighting durable disease control. The confirmed objective response rate (ORR) was 2.3% [101].

In lung and thymic NETs, the randomized phase II LUNA trial evaluated pasireotide, everolimus, and their combination. The highest 9-month PFS rate (58.5%) was observed in the combination arm, outperforming monotherapy with pasireotide (39%, 95% CI 24.2–55.5) and everolimus (33.3%, 95% CI 19.6–49.5) [102].

Emerging evidence also suggests that metabolic targeting with agents like metformin may enhance the efficacy of SSA-mTOR inhibitor combinations. A retrospective analysis by Pusceddu et al. found significantly longer mPFS (29 vs. 11 months; *p* = 0.018) in diabetic patients receiving everolimus, SSA, and metformin compared to non-diabetics without metformin [103,104]. Building on these findings, the MetNet trial (NCT02294006) is currently evaluating this three-drug regimen in advanced panNETs [105].

#### 5.2.2. SSAs and Peptide Receptor Radionuclide Therapy (PRRT)

SSAs also play a crucial role in optimizing PRRT. By upregulating SSTR expression and improving radiolabeled peptide retention, SSAs may enhance the therapeutic efficacy of PRRT [90]. This synergistic relationship underscores the importance of careful treatment sequencing and the potential for combination strategies to maximize patient outcomes [23].

The antitumor activity and safety of the combined therapy of SSAs with 177Lu-DOTA0-Tyr3-Octreotate (Luthatera^®^) have been demonstrated in GEP-NETs by NETTER-1 and NETTER-2 trials [106,107,108]. The approval of Lutathera^®^ by the European Medicines Agency (EMA) in 2017 and the US Food and Drug Administration (FDA) in 2018 as a second-line strategy for unresectable, metastatic, or locally advanced, G1-G2, SSTR-positive midgut NETs in progression on SSAs was remarkably supported by the phase-3 NETTER-1 trial [106,108]. Patients treated with Lutathera^®^ combined with octreotide 30 mg/28 days, compared to high-dose octreotide (60 mg/28 days), significantly improved PFS, response rate, and quality of life. Furthermore, NETTER-2 confirmed the relevant benefits in survival and tumor response of Luthatera^®^ combined with octreotide LAR 30 mg/28 days compared to octreotide LAR 60 mg/28 days alone (control arm), as first-line treatment in advanced, higher grade 2 (Ki67 ≥ 10% and ≤20%) and grade 3 (Ki67 > 20% and ≤55%), SSTR-positive GEP-NETs. mPFS was significantly higher in the Luthatera^®^ arm versus the control arm (22.8 months vs. 8.5 months, stratified hazard ratio 0.276 [109]; *p* < 0.0001), reducing the risk of disease progression or death by 72% compared with the control arm. ORR was superior in the treatment arm compared with the control arm (43.0% vs. 9.3%, respectively) [107]. Despite the clear benefit in favor of PRRT compared to high doses of SSA, 8.5 months of PFS in this category of well-differentiated neoplasms with high ki67 remains, at the moment, an astounding and unexpected finding. This result underlines how even high doses of SSA must still be explored in dedicated studies before decreeing their usefulness.

### 5.3. High-Dose SSAs and Emerging Formulations

High-dose SSAs have emerged as a promising therapeutic strategy for patients with progressive GEP-NETs following standard-dose treatments.

A phase 2 trial investigating high-dose regimens in GEP-NETs in progression after SSAs at standard doses highlighted that an increased frequency of administration significantly impacts PFS in patients with Ki67 index ≤10%, while no worsening of the safety profile was observed [110]. Despite limited clinical trials, real-world studies have reported clinical benefits, particularly in refractory hormonal syndromes where high-dose SSAs restored biochemical and symptomatic control. A retrospective analysis by Lamberti et al. further supports the utility of high-dose SSAs in patients with advanced, well-differentiated GEP-NETs who experienced disease progression under conventional dosing [111]. In this study, high-dose SSAs were administered in either increased frequency or double doses. The results demonstrated a significant extension of median PFS (19.6 months vs. 9.7 months; *p* < 0.001) in patients receiving intensified SSA regimens compared to standard dosing. Additionally, 60% of patients achieved disease stabilization, reinforcing the cytostatic potential of high-dose SSAs without notable increases in adverse effects. These findings highlight the role of intensified SSA therapy as an effective salvage approach in progressive cases and emphasize the need for prospective trials to define its optimal use. Additionally, there is evidence supporting tumor growth inhibition, especially in low-grade tumors [56]. However, findings from a recent meta-analysis, which synthesized data from heterogeneous studies primarily retrospective in nature, suggest that while high-dose SSAs show potential as a salvage approach, their efficacy in preventing disease progression is limited in unresectable GEP-NETs, and the use of this therapeutic approach is advisable in selected cases when other antiproliferative treatments are not feasible [112]. However, clinical evidence consistently demonstrates that SSAs maintain a favorable safety profile even at high doses, with no additional side effects reported at the highest doses used in clinical practice. Tailored dose regimens are critical to optimizing therapeutic outcomes while preserving this safety profile. Moreover, combination therapies with other targeted treatments offer an avenue to enhance efficacy without increasing the dependency on high-dose SSAs [110].

The high variability across studies underscores the need for prospective trials to define optimal use. Moreover, the optimal positioning of high-dose SSAs remains uncertain, necessitating further research to validate findings and establish dosing protocols.

Novel formulations, including oral and subcutaneous SSAs, represent significant advancements aimed at improving pharmacokinetics, patient compliance, and treatment convenience (Table 4).

Oral octreotide capsules represent a promising development. Studies like the MPOWERED phase III core trial [113] and open-label extension [109] in acromegaly demonstrated that oral octreotide effectively reduced dependence on injections, improving adherence and quality of life. Although its application in NENs is still under investigation, oral formulations have the potential to expand therapeutic options significantly.

Another novel drug is paltusotine (previously known as CRN00808), a once-daily, nonpeptide, selective SSTR2 agonist, which has already been shown to ameliorate the biochemical and clinical control of acromegalic individuals in phase II and III trials with a comparable safety profile to the injectable SSTR ligands [114,115]. Currently, a randomized phase II study (NCT05361668) is evaluating paltusotine in locally advanced or metastatic NET patients with carcinoid syndrome. The primary endpoint focuses on safety and dose–response assessment, highlighting its potential for broader application. A subcutaneous octreotide Depot Formulation (CAM2029) is another formulation explored both in acromegalic and NET patients. CAM2029 is a long-acting, high-exposure octreotide formulation, which offers better bioavailability and easier administration compared to octreotide LAR [116]. A phase II, randomized, open-label, multicenter trial evaluated its efficacy and safety in patients with acromegaly and functioning NETs (ileum 80% and ileocecum 20%) previously treated for at least 2 months with octreotide LAR i.m. 10–20–30 mg/month. Twelve subjects were randomized to octreotide 10 mg s.c. every 2 weeks (NET n = 1, acromegaly n = 3) or 20 mg s.c. every 4 weeks (NET n = 4, acromegaly n = 4) for 3 months. Results demonstrated bioavailability superiority and maintained or improved symptom control in NET patients, along with stabilization or enhancement of IGF-1 and GH levels in acromegalic patients. The safety findings were consistent with the known toxicity profile of octreotide i.m. [117].

Regarding NETs, the SORENTO trial (NCT05050942), an ongoing phase III randomized, open-label, multicenter trial, aims to evaluate the efficacy and safety of CAM2029 (20 mg s.c. twice weekly) compared to lanreotide ATG (120 mg/month) or octreotide LAR (30 mg/month) in advanced, well-differentiated GEP-NETs. Primary endpoints include PFS, with secondary evaluations of OS, ORR, and safety profiles over six years.

## 6. Conclusions

SSAs remain a cornerstone of NEN therapy due to their antiproliferative and hormonal control properties. Despite advancements, challenges such as receptor downregulation and tumor heterogeneity necessitate innovative approaches. Emerging high-dose formulations, oral alternatives, and combination therapies with PRRT or mTOR inhibitors show promising efficacy (Figure 2).

Yet several open issues persist regarding their long-term use, eventual discontinuation, sequencing, combination regimens, and integration with other modalities. Ongoing trials like SAUNA and SANO are expected to provide insights into long-term benefits and maintenance strategies. Innovations in SSA formulations, dosing strategies, and combination approaches offer a promising path forward. Ultimately, addressing these questions will require robust clinical trials and multidisciplinary collaboration. By addressing these gaps in knowledge, the therapeutic potential of SSAs can be fully realized, paving the way for more effective and personalized approaches in the management of neuroendocrine neoplasms.

## Figures and Tables

**Figure 1 cells-14-00245-f001:**
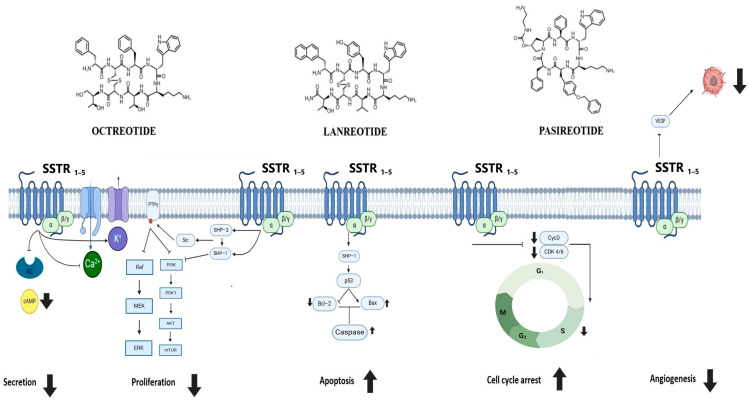
Molecular mechanisms and effects of somatostatin analogs (SSAs) summarized.

**Figure 2 cells-14-00245-f002:**
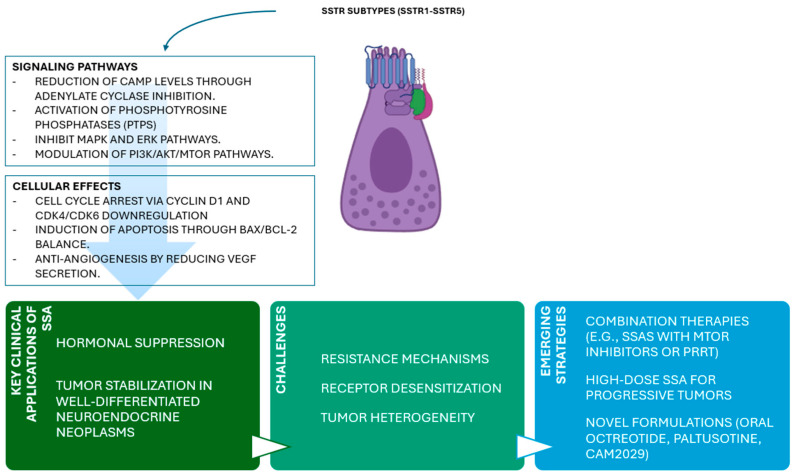
Cellular and molecular mechanisms of somatostatin analogs (SSAs), with their therapeutic applications, challenges, and future directions.

**Table 1 cells-14-00245-t001:** Distribution and functions of somatostatin receptors (SSTRs).

SSTR Subtype	Primary Tissue Distribution	Major Physiological Functions
SSTR1	CNS, GI tract, kidneys	Neuronal signaling, GI motility regulation
SSTR2	CNS, endocrine pancreas, GI tract	Hormone secretion inhibition, antiproliferative effects
SSTR3	Pancreas, lungs, brain	Cell cycle arrest, apoptosis
SSTR4	Brain, lungs	Neurotransmission modulation
SSTR5	Pituitary, pancreas, GI tract	Regulation of growth hormone and insulin secretion

CNS = central nervous system, GI tract = gastrointestinal tract.

**Table 2 cells-14-00245-t002:** Hormone-suppressive and antiproliferative effects of somatostatin analogs (SSAs).

Mechanisms	Effects
Hormone-Suppressive Effects	
Inhibition of cAMP signaling	Reduces hormone secretion through SSTR2 and SSTR5 interaction
Pituitary	Suppresses GH, TSH
Pancreas	Suppresses Insulin, Glucagon, VIP, PP
Gastrointestinal	Suppresses Gastrin, Serotonin, VIP
Antiproliferative Effects	
Activation of phosphotyrosine phosphatases (PTPs)	Inhibits tyrosine kinases and mitogenic pathways like MAPK and ERK, suppressing growth
Modulation of PI3K/AKT/mTOR pathway	Promotes apoptosis and inhibits tumor growth
Cell cycle arrest (G1 phase)	Halts cell division by downregulating cyclins and CDKs
Pro-apoptotic effects	Alters Bax/Bcl-2 ratio, promoting caspase activation and apoptosis
Anti-angiogenic properties	Reduces VEGF and impairs angiogenesis, limiting tumor vascularization
Modulation of tumor microenvironment	Reduces inflammatory signals and immune suppression, impairing tumor progression

TSH: thyroid-stimulating hormone, GH: growth hormone, VIP: vasoactive intestinal peptide, PP: pancreatic polypeptide, cAMP: cyclic adenosine monophosphate, PTPs: phosphotyrosine phosphatases, MAPK: mitogen-activated protein kinase, ERK: extracellular signal-regulated kinase, PI3K/AKT/mTOR: phosphatidylinositol 3-kinase/protein kinase B/mammalian target of rapamycin, CDKs: cyclin-dependent kinases, Bax: Bcl-2-associated X protein, Bcl-2: B-cell lymphoma 2, VEGF: vascular endothelial growth factor.

**Table 3 cells-14-00245-t003:** Pathological classification and SSA responsiveness of neuroendocrine tumors.

Tumor Type	WHO Grade	SSTR Expression	SSA Responsiveness	Characteristics
GI NETs	Usually WD G1–G3	High	High	Includes stomach, duodenum, small intestine, and rectum NETs
Pancreatic NENs	WD G1–G3 (NET) PD (NEC)	Moderate to High Low	Variable Low	Functioning and non-functioning NENs Ki67 high/very high (typically > 55%); aggressive growth
Lung NENs	Typical/Atypical (NET) Small and Large Cell (NEC)	Variable Low	Low to Moderate Minimal	Typical and atypical carcinoids Poorly differentiated, small and large cell NEC; aggressive growth

NET, neuroendocrine tumor; NEC, neuroendocrine carcinoma; WD: well-differentiated; PD: poorly differentiated.

**Table 4 cells-14-00245-t004:** Recent clinical trials on novel SSA formulations.

Formulation	Study/Trial	Phase	Indication	Key Outcomes
Oral Octreotide	MPOWERED (NCT02685709)	Phase III	Acromegaly	Demonstrated effective reduction in injection dependence; improved adherence and quality of life.
	Open-label extension	Phase III	Acromegaly	Maintained biochemical control with improved patient satisfaction.
Paltusotine	NCT05361668	Phase II/III	NETs with carcinoid syndrome	Safety and dose–response assessment underway, showing potential broad application.
CAM2029	NCT02299089	Phase II	Functioning NETs and acromegaly	Superior bioavailability and efficacy compared to octreotide LAR; stable symptom control.
	SORENTO (NCT05050942)	Phase III	Advanced GEP-NETs	Ongoing evaluation of PFS and safety compared to standard therapies over six years.

## Data Availability

No new data were created or analyzed in this study. Data sharing is not applicable to this article.

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
