# Peer review of "“Cold” Somatostatin Analogs in Neuroendocrine Neoplasms: Decoding Mechanisms, Overcoming Resistance, and Shaping the Future of Therapy"

_cells, 2025, doi:10.3390/cells14040245_

Round 1

Reviewer 1 Report

Comments and Suggestions for Authors

It was great to read this interesting manuscript and thanks to the authors for shedding more light on the NETs subject.

My comments:

1. Paragraph 87-99: 68Ga-DOTATATE is indeed the most commonly used imaging modality for NETs however, 18F-NOTA-Octreotide has revolutionized the imaging space recently. I suggest authors include 18F-NOTA-Octreotide in paragraph 87-99. Authors can get inspiration from this paper: https://jnm.snmjournals.org/content/64/6/835.

2. Paragraph 187: Do all SSAs binding to SSTRs undergo cell internalization? Is there antagonist binding activity reported in literature?

3. Will dose escalation not rather saturate the receptors or even cause side effects. The receptors as mentioned are not only expressed on the cancer cells.

4. Paragraph 214: I suggest authors include 18F-NOTA-Octreotide. This modality is currently preferred over 68Ga-DOTATATE.

5. I agree with your future directions. Currently, the most commonly used radioactive agent is 177Lu which is beta emitter and can only induce single strand DNA break. Will authors agree that alpha emitters which are able to cause double strand DNA break will increase the potency and cell killing effect of SSAs like 225Ac and 213Bi. Could authors include this in the future direction? Authors can be inspired from this paper: https://www.mdpi.com/1999-4923/13/5/599.

Author Response

We sincerely thank the reviewer for his thoughtful comments and valuable insights on the subject of neuroendocrine tumors (NETs). Your expertise and constructive feedback have greatly contributed to improving the quality and clarity of our work.

Comments:

  1. Paragraph 87-99: 68Ga-DOTATATE is indeed the most commonly used imaging modality for NETs however, 18F-NOTA-Octreotide has revolutionized the imaging space recently. I suggest authors include 18F-NOTA-Octreotide in paragraph 87-99. Authors can get inspiration from this paper: https://jnm.snmjournals.org/content/64/6/835.

We appreciate the reviewer’s comment regarding 18F-NOTA-Octreotide. The manuscript has been updated to include a discussion on this novel imaging modality. We highlighted its advantages over 68Ga-DOTATATE, such as enhanced image resolution, longer half-life, and broader accessibility due to cyclotron production. The recommended reference has been cited to provide a comprehensive overview of its role in NET imaging.

  1. Paragraph 187: Do all SSAs binding to SSTRs undergo cell internalization? Is there antagonist binding activity reported in literature?

Thank you for raising this point. The manuscript now clarifies that while most SSAs binding to SSTRs induce receptor internalization, the degree of this process depends on the specific receptor subtype and ligand. Additionally, we included a discussion on SSTR antagonists, which do not trigger internalization but retain the receptors on the cell surface, thereby potentially enhancing their use in radiolabeled therapies.

  1. Will dose escalation not rather saturate the receptors or even cause side effects. The receptors as mentioned are not only expressed on the cancer cells.

We have addressed the concern regarding receptor saturation and side effects of dose escalation in the revised manuscript. The updated text explains the potential risks of saturating receptors expressed on both tumor and normal tissues. However, we also emphasized that SSAs, even at high doses, have demonstrated a favorable safety profile, with no additional side effects reported even at the highest doses used in clinical practice. Furthermore, we included strategies such as tailored dose regimens and the use of combination therapies to optimize therapeutic efficacy while maintaining this excellent safety standard.

  1. Paragraph 214: I suggest authors include 18F-NOTA-Octreotide. This modality is currently preferred over 68Ga-DOTATATE.

The manuscript now reflects the growing preference for 18F-NOTA-Octreotide over 68Ga-DOTATATE in clinical practice. We emphasized its superior technical and clinical advantages, aligning with current trends in NET imaging.

  1. I agree with your future directions. Currently, the most commonly used radioactive agent is 177Lu which is beta emitter and can only induce single strand DNA break. Will authors agree that alpha emitters which are able to cause double strand DNA break will increase the potency and cell killing effect of SSAs like 225Ac and 213 Could authors include this in the future direction? Authors can be inspired from this paper: https://www.mdpi.com/1999-4923/13/5/599.

In response to the reviewer’s insightful comment, we included a section in the future directions highlighting the potential of alpha emitters such as 225Ac and 213Bi. Their ability to cause double-strand DNA breaks significantly enhances cytotoxicity compared to beta emitters. The suggested reference has been cited to support this addition, and we discussed how integrating alpha emitters into SSA-based therapies may further improve treatment outcomes for NETs.

Reviewer 2 Report

Comments and Suggestions for Authors

Excellent paper on an important topic given that it is sharply increasing in the population and we have noticed this greater presence in the last 15 years. Previously, the generations of the sixth and seventh decade of life were affected, now they are demonstrated even in young ages, proof of this is a beautiful article (doi.org/10.3390/cancers16203440 to read and cite in bibliography) very recent. Excellent introduction on which we agree on everything, in fact we believe that somatostatin and analogues are still the cornerstone of therapy for neuroendocrine tumors even if it must always be kept in mind that it can cause gallstones, which therefore surgery is recommended to remove (PMID: 38051513 to read and cite in bibliography). The dissertation on SSTR receptors can be defined as learned, although not everyone knows it. Also important is the emphasis on somatostatin analogues that "promote cellular apoptosis. We absolutely agree with the synergy of everolimus with somatostatin analogues. Corì as the minor efficacy on the less differentiated forms is described. Excellent overview of the possibilities that the industry offers us. Correct conclusion arising from the broad and fair discussion. Good English, good iconography, good bibliography

Author Response

We greatly appreciate the reviewer’s thorough evaluation and positive feedback on our manuscript. Below is our point-by-point response to the comments:

  1. Observation of Neuroendocrine Tumor (NET) Trends Over Time. Previously, the generations of the sixth and seventh decade of life were affected, now they are demonstrated even in young ages, proof of this is a beautiful article (doi.org/10.3390/cancers16203440 to read and cite in bibliography) very recent.

Response: We have noted and included in the introduction that the incidence of NETs is not only increasing but is also being observed in younger populations compared to earlier decades. We have cited the suggested reference (doi.org/10.3390/cancers16203440) to support this observation and to provide a comprehensive context regarding the epidemiological trends in NETs.

  1. Gallstones as a Complication of Somatostatin Analogues (SSAs). "It must always be kept in mind that it can cause gallstones, which therefore surgery is recommended to remove (PMID: 38051513 to read and cite in bibliography)."

Response: The comment regarding the potential for gallstones as a complication of SSA therapy and the recommendation for surgical intervention is well-taken. We have updated the discussion on SSA therapy to acknowledge this side effect (see Resistance Mechanisms and Limitations section), emphasizing its clinical relevance. The recommended reference (PMID: 38051513) has been included.

We are grateful for the reviewer’s positive assessment of the manuscript’s conclusion, English quality, iconography, and bibliography. The inclusion of the references suggested further strengthens the scientific rigor of our work.

Reviewer 3 Report

Comments and Suggestions for Authors

Cells-3448834: “Cold” Somatostatin Analogs in Neuroendocrine Neoplasms: Decoding Mechanisms, Overcoming Resistance, and Shaping the Future of Therapy by Sara Massironi et al.

General Comments:

The authors reviewed on the neuroendocrine neoplasms (NENs) and the developmental progress of usage of somatostatin analogs (SSAs) for the NEN treatment by targeting somatostatin receptors (SSTRs).  This review explores the cellular and molecular mechanisms underlying the antitumor effects of SSAs, including receptor-mediated signaling pathways, cell cycle arrest, apoptosis induction, and anti-angiogenesis.  This also handles the role of SSAs in combination therapies with mTOR inhibitors, peptide receptor radionuclide therapy (PRRT), and novel formulations, including oral derivatives and subcutaneous depot formulations.  The review paper provides a comprehensive analysis of the cellular and molecular mechanisms of SSAs, although elucidation of the resistance mechanisms and the approaches to overcome the resistance have yet to be completed.  This review paper includes a wide range of the clinical significance of SSTRs and the utility of SSAs; however, some issues are raised to improve as a comprehensive review.

Specific Comments:

1) The normal distribution of SSTRs and their function in each tissue and organs should be addressed before the section of SSTRs in the NET tissues. The referee recommends to put a figure or a table regarding the normal SSTR distribution and the major functions.

2) SSAs exert both antisecretory and antiproliferative properties, achieving their effects through binding to somatostatin receptors (SSTRs) expressed on NET tissues.  Therefore, Table 1 should be separated to show the anti-proliferative effects and hormone-suppressive effects individually, since these mechanisms in each tissue may be diverged. 

3) In the section of the heterogeneity of the tumors, it should be needed to contain the pathological classification of NET/NEC and their characteristics and the individual tissues of NETs such as GI, pancreatic and lung NETs.  The referee recommends to add a new table to show the pathologic aspect on the effects and tolerances of SSAs,

4) The recent trials including oral derivatives and other formulas, and the related clinical trials should be shown in a separated table, since various trials are included in this review. 

5) The elucidation of the mechanisms of SSAs resistance and the approaches to overcome the resistance need to be discussed in detail.  In the case of pit-NETs such as acromegaly and Cushing diseases, the SSA resistance are less observed compared with other NETs such as GI, pancreatic and lung NETs.  The differences of not only pathological grade but also tissue differences should also be involved in this mechanism.

6) Also, the possible mechanism in which the combination therapy of SSAs and mTOR inhibitors can be applied for the refractory NET should be explained.

Author Response

Reviewer 3

We greatly appreciate the reviewer’s thorough evaluation and positive feedback on our manuscript. Below our point-to-point responses

  1. The normal distribution of SSTRs and their function in each tissue and organs should be addressed before the section of SSTRs in the NET tissues. The referee recommends putting a figure or a table regarding the normal SSTR distribution and the major functions. Response: We agree with the reviewer and we have added a new subsection before the “SSTRs in NET Tissues” section, describing the normal distribution of SSTRs and their major physiological functions. Additionally, we have included a new table (Table 1) summarizing the distribution of SSTR subtypes across different tissues and their associated functions.
  1. SSAs exert both antisecretory and antiproliferative properties, achieving their effects through binding to somatostatin receptors (SSTRs) expressed on NET tissues. Therefore, Table 1 should be separated to show the anti-proliferative effects and hormone-suppressive effects individually, since these mechanisms in each tissue may be diverged.
    Response:
    We appreciate the reviewer’s suggestion. To address this, we have separated Table (now 2) into two distinct subsections: one focusing on the hormone-suppressive effects of SSAs and the other on their antiproliferative effects. Each part now highlights the tissue-specific mechanisms and outcomes, providing clearer insight into these divergent effects.
  2. In the section of the heterogeneity of the tumors, it should be needed to contain the pathological classification of NET/NEC and their characteristics and the individual tissues of NETs such as GI, pancreatic, and lung NETs. The referee recommends adding a new table to show the pathological aspects of the effects and tolerances of SSAs.
    Response:
    We have expanded the “Heterogeneity of Tumors” section to include emphasis on the tumors characteristics and expression of SSTRs. While the WHO grading system is already introduced in the manuscript's introduction, we now added the variability in SSTR expression among different NET categories, which directly impacts their responsiveness to SSAs. Furthermore, we have added a new table (Table 3), summarizing the pathological aspects, SSTR expression, and SSA responsiveness for GI NETs, pancreatic NETs, lung NETs, and NECs.
  3. The recent trials including oral derivatives and other formulas, and the related clinical trials should be shown in a separate table, since various trials are included in this review.
    Response:
    We have created a new table (Table 4) listing recent clinical trials involving oral SSA derivatives, subcutaneous depot formulations, and other innovative SSA formulations.
  4. The elucidation of the mechanisms of SSA resistance and the approaches to overcome the resistance need to be discussed in detail. In the case of pit-NETs such as acromegaly and Cushing diseases, the SSA resistance is less observed compared with other NETs such as GI, pancreatic, and lung NETs. The differences in not only pathological grade but also tissue differences should also be involved in this mechanism.
    Response: We thank the reviewer for their insightful comments regarding SSA resistance mechanisms. in response to the reviewer’s suggestion, we have briefly addressed the distinct resistance mechanisms observed in pit-NETs (see tumor heterogeneity), even if pituitary NETs (pit-NETs) are actually outside the primary scope of this manuscript.
  5. Also, the possible mechanism in which the combination therapy of SSAs and mTOR inhibitors can be applied for refractory NET should be explained.
    Response:
    We have included a detailed discussion on the synergistic mechanisms of SSAs and mTOR inhibitors in refractory NETs (see 5.2.1. SSAs and mTOR Inhibitors). This section elaborates on how mTOR inhibitors may enhance SSA efficacy by targeting downstream signaling pathways and overcoming resistance mechanisms. We have also cited relevant preclinical and clinical studies to support this discussion.

We hope these revisions adequately address the reviewer’s concerns and improve the clarity and comprehensiveness of the manuscript. We are grateful for these insightful comments and welcome any further suggestions.

Round 2

Reviewer 3 Report

Comments and Suggestions for Authors

The authors responded to the referees' suggestions and appropriately revised their review manuscript.  The referee would like to recommend to add a summarized figure that comprehensively indicates cellular and molecular mechanisms of SSAs and the future application of SSAs for the readers.

Author Response

Dear Reviewer,

We would like to express our gratitude for your thorough review and valuable suggestions, which have significantly improved our manuscript.

Comment 1: The referee would like to recommend adding a summarized figure that comprehensively indicates cellular and molecular mechanisms of SSAs and the future application of SSAs for the readers.

Response: We appreciate this excellent suggestion, which enhances the clarity and utility of our review. We have included a newly created figure (Figure 2) that summarizes the cellular and molecular mechanisms of somatostatin analogs (SSAs) and highlights their current therapeutic applications as well as future directions.